# Targeting Chondrocyte Hypertrophy as Strategies for the Treatment of Osteoarthritis

**DOI:** 10.3390/bioengineering12010077

**Published:** 2025-01-15

**Authors:** Da-Long Dong, Guang-Zhen Jin

**Affiliations:** 1Institute of Tissue Regeneration Engineering (ITREN), Dankook University, Cheonan 31116, Republic of Korea; dongdalong@dankook.ac.kr; 2Department of Nanobiomedical Science and BK21 PLUS NBM Global Research Center for Regenerative Medicine, Dankook University, Cheonan 31116, Republic of Korea

**Keywords:** osteoarthritis, chondrocyte hypertrophy, signaling pathways, small molecule compounds, tissue engineering

## Abstract

Osteoarthritis (OA) is a common joint disease characterized by pain and functional impairment, which severely impacts the quality of life of middle-aged and elderly individuals. During normal bone development, chondrocyte hypertrophy is a natural physiological process. However, in the progression of OA, chondrocyte hypertrophy becomes one of its key pathological features. Although there is no definitive evidence to date confirming that chondrocyte hypertrophy is the direct cause of OA, substantial experimental data indicate that it plays an important role in the disease’s pathogenesis. In this review, we first explore the mechanisms underlying chondrocyte hypertrophy in OA and offer new insights. We then propose strategies for inhibiting chondrocyte hypertrophy from the perspectives of targeting signaling pathways and tissue engineering, ultimately envisioning the future prospects of OA treatment.

## 1. Introduction

Articular cartilage primarily refers to hyaline cartilage, with a thickness of 2 to 4 mm, composed of chondrocytes (~3–5%) and an extracellular matrix (ECM) [1,2]. Chondrocytes play a crucial role in maintaining the health and function of articular cartilage, while the ECM consists of water (~65–75%), collagen (~20–30%), and proteoglycans (~10%). The collagen is primarily composed of types II, IX, X, and XI, with type II accounting for over 90%. Chondrocytes in the superficial zone mainly synthesize type II collagen, while hypertrophic chondrocytes in the deeper ossification zone actively produce type X collagen [3,4]. Although normal chondrocytes are quiescent cells, they are responsible for ECM synthesis, and maintain cartilage homeostasis under normal physiological conditions by balancing anabolic and catabolic processes [5].

The ECM is crucial for maintaining the structural integrity and elasticity of cartilage. In a healthy joint, the ECM provides mechanical support to the chondrocytes embedded within it, and facilitates smooth joint movement. These chondrocytes respond to mechanical loads and biochemical signals [6,7]. For example, the expression of the key transcription factor SOX9, which regulates ECM genes (such as collagen, hyaluronic acid, and proteoglycans) during cartilage development, is modulated by mechanical stimulation and growth factors like TGF-β [8]. Additionally, interactions between chondrocytes and the surrounding ECM are vital for cell survival and function, as these interactions influence signaling pathways that regulate cell proliferation, differentiation, and matrix production [9].

Cartilage is an avascular, alymphatic, and aneural connective tissue. Chondrocytes obtain nutrients through diffusion from the synovial fluid at the joint surface. As a result, the deeper layers of cartilage exist in a low-oxygen environment, leading to a low turnover rate of chondrocytes [10,11]. Furthermore, due to the relatively weak regenerative capacity of cartilage, once damage occurs for various reasons, it is difficult for cartilage to repair itself [12].

Osteoarthritis (OA) is a globally prevalent chronic degenerative joint disease characterized by the gradual degeneration of articular cartilage, leading to pain, stiffness, and loss of function in the affected joints. The incidence is notably higher in women, especially among the elderly, and is associated with various risk factors, including age, obesity, and joint injuries [13]. The pathophysiology of OA involves complex interactions between mechanical stress, inflammatory processes, and metabolic changes, all of which contribute to cartilage degradation and joint dysfunction [14]. Numerous studies indicate that biological factors, including proteases, pro-inflammatory cytokines, and reactive oxygen species (ROS), accelerate the progression of OA [15]. Proteases include matrix metalloproteinases (MMPs) and a disintegrin and metalloproteinase with thrombospondin motifs (ADAMTS) [16]. Pro-inflammatory cytokines such as IL-1β, IL-6, and TNF-α are associated with the inflammatory response triggered by degradation products of the cartilage matrix [17,18]. ROS production is closely related to the upregulation of pro-inflammatory cytokines and oxidative stress caused by mechanical stress [19,20,21]. These factors not only act independently, but also interact with each other, accelerating the progression of arthritis. Furthermore, growing evidence shows a strong link between metabolic syndrome and OA. Obesity not only increases mechanical stress on the joints, but also promotes systemic inflammation, further worsening OA symptoms [22]. Although chondrocyte hypertrophy is a hallmark of OA, a direct causal link between them remains unconfirmed. This review will discuss the underlying mechanisms of chondrocyte hypertrophy in OA, and explores potential strategies to mitigate its effects.

## 2. OA and Chondrocyte Hypertrophy

Chondrocyte hypertrophy is a natural physiological phenomenon that occurs during bone development. The initial phase of bone development is the formation of temporary cartilage, which involves the aggregation of mesenchymal stem cells (MSCs) and their differentiation into chondrocytes. During this stage, chondrocytes actively proliferate, accompanied by the expression of SOX9 and type II collagen. This process then progresses to endochondral ossification, where chondrocytes undergo hypertrophic differentiation and the surrounding tissue begins to mineralize, ultimately contributing to the remodeling of cartilage into bone tissue. At this stage, the expression of Indian Hedgehog (IHH), type X collagen, RUNX2, alkaline phosphatase (ALP), and osteopontin (OPN) is significantly elevated [23]. Throughout bone development, hypertrophic chondrocytes face three possible fates: apoptosis, transdifferentiation into osteoblast-like cells, and senescence [24,25]. The senescence of chondrocytes occurs through one of two mechanisms—replicative senescence and stress-induced premature senescence. The p53/p21 pathway induces replicative senescence, while the p38/p16 pathway induces stress-induced senescence [26,27]. Senescent chondrocytes are characterized by increased senescence-associated β-galactosidase activity [28], cell growth arrest, the production of senescence-associated secretory phenotypes (SASP) [29,30], and a reduction in chondrocyte-specific proteins such as type II collagen and proteoglycans [31].

In OA, a phenomenon similar to chondrocyte hypertrophy observed during normal bone development is present, along with an increase in proteases and inflammatory cytokines, collectively referred to as the three major pathological features of OA. At this stage, quiescent chondrocytes begin to actively proliferate, initiating chondrocyte hypertrophy, which results in the loss of their differentiated phenotype (type II collagen, aggrecan) and the expression of typical hypertrophic markers such as type X collagen, RUNX2, MMP-13, and ADAMTS-5 [32,33]. In OA, chondrocytes exhibit a hypertrophic phenotype that is specifically related to the terminal stage of chondrocyte differentiation. Under normal circumstances, the terminal differentiation of chondrocytes in joints is suppressed, while OA joint cartilage contains an excessive number of terminally differentiated chondrocytes [34]. MMP-13 and type X collagen are key markers of the mechanism of OA caused by chondrocyte hypertrophy. As a representative transcription factor of chondrocyte hypertrophy, RUNX2 promotes the terminal differentiation of chondrocytes and upregulates the expression of type X collagen [35,36]. Additionally, C/EBPβ and hypoxia-inducible factor-2α (HIF-2α) play important roles in chondrocyte terminal differentiation. C/EBPβ is a crucial transcription factor that induces chondrocyte hypertrophy [37,38], while HIF-2α not only upregulates the expression of type X collagen, MMP-13, and VEGF, but also acts as an inducer of C/EBPβ, thereby promoting cartilage fibrosis and vascularization. Numerous studies using animal models have shown a close correlation between these factors, the regulation of chondrocyte hypertrophy, and OA. Research indicates that inhibiting transcription factors such as RUNX2, C/EBPβ, and HIF-2α that promote chondrocyte hypertrophy can effectively protect mice from the progression of OA [37,39,40]. Wnt, BMP, and TGF-β signaling pathways are the main regulatory pathways associated with chondrocyte hypertrophy, and they can induce hypertrophy and are closely related to OA development [34,41]. In OA, chondrocyte hypertrophy accelerates the activation of RUNX2, promotes the terminal differentiation of chondrocytes, increases the expression of type X collagen and ALP, and ultimately leads to the calcification of the cartilage matrix [42,43,44]. Therefore, targeting and inhibiting chondrocyte hypertrophy may be a promising strategy for the treatment of OA. Therefore, targeting and inhibiting chondrocyte hypertrophy could be a promising strategy for treating OA.

## 3. Primary Cellular Signaling Pathways in Chondrocyte Hypertrophy and Their Corresponding Therapeutic Strategies

In OA, quiescent articular chondrocytes transition into active proliferation and gradually advance toward hypertrophic differentiation. This process is accompanied by cartilage matrix degradation, the downregulation of cartilage-specific markers, and an increase in bone markers. A complex network of signaling pathways regulates these events. Here, we focus our discussion on the signaling pathways that regulate articular chondrocyte hypertrophy, including IHH/PTHrP, TGF-β, BMP, Wnt, FGF, HIF-2α, NF-κB, VEGF, YAP, and mTOR (Figure 1).

### 3.1. IHH/PTHrP Signaling

The IHH/PTHrP signaling pathway plays a crucial role in the phenotypic formation of chondrocytes in the growth plate and is a key regulatory factor in endochondral ossification [45]. IHH is expressed and secreted by prehypertrophic chondrocytes, where it binds to the Patched 1 (PTCH1) receptor, thereby counteracting the inhibition caused by the Smoothened (SMO) protein and activating downstream transcription factors GLI2. These transcription factors translocate to the nucleus and enhance the expression of hypertrophic markers such as type X collagen and MMP-13. This process is closely associated with the progression of OA [39,40]. Parathyroid hormone-related protein (PTHrP) binds to the G protein-coupled receptor PTH1R, and inhibits the hypertrophic phenotype of chondrocytes by activating the adenylyl cyclase/protein kinase A (PKA) pathway and inducing the expression of BAPX1/NKX3.2 [46,47,48]. PTHrP also plays an essential role in regulating the IHH signaling pathway, primarily by delaying chondrocyte proliferation and inhibiting hypertrophy [41]. IHH regulates the expression of PTHrP, while PTHrP provides negative feedback on IHH, with both factors working together to coordinate the hypertrophic differentiation of chondrocytes. An overview of the role of the Ihh/PTHrP signaling pathway in regulating chondrocyte hypertrophy is provided in Figure 1.

Chang et al. found that the intra-articular injection of PTHrP in a chemically induced rat OA model reduces the loss of proteoglycans, increases the level of type II collagen, and decreases the expression of type X collagen, thereby alleviating the progression of OA [49]. Another study incorporated PTHrP into a drug delivery system, which reduced proteoglycan loss in a chemically induced mouse OA model, accompanied by reduced type X collagen staining. PTHrP demonstrated a protective anti-hypertrophic effect, reducing the severity of OA [50]. Consistent with the important role of Ihh in chondrocyte hypertrophy, the expression of IHH in OA cartilage and synovial fluid is significantly increased compared to non-OA cartilage. When the Ihh signaling pathway is inhibited by cyclopamine, the expression of chondrocyte hypertrophy markers decreases [51]. Furthermore, in the study by Zhou et al., Ihh gene knockout mice largely prevented the development of traumatic OA.

Small molecule compounds, including agonists and antagonists of cellular signaling pathways, have shown considerable promise for targeting specific sites that trigger biological effects, particularly in inhibiting chondrocyte hypertrophy in OA. Several therapeutic strategies have been proposed to counteract the abnormal activation of the IHH/PTHrP signaling pathway in OA. For instance, inhibiting IHH signaling can alleviate hypertrophic changes in OA cartilage and support cartilage matrix synthesis. Zhou et al. demonstrated the therapeutic potential of Cyclopamine, an IHH inhibitor, through in vitro and in vivo experiments involving OA patient-derived chondrocytes and a mouse model. They found that Cyclopamine effectively reduced the expression of hypertrophic markers such as type X collagen and MMP-13, significantly slowing OA progression in mice [52]. Similarly, Guo et al. tested another IHH inhibitor, ipriflavone, in a rat model, and observed a reduction in hypertrophic markers (e.g., type X collagen and MMP-13) while promoting cartilage phenotype markers like type II collagen and proteoglycans. These effects were attributed to the inhibition of SMO and GLI signaling molecules within the IHH pathway [53]. Additionally, intermittent PTHrP administration has been shown to enhance chondrogenesis in MSCs and reduce hypertrophy [54]. These findings highlight the importance of modulating the IHH/PTHrP signaling pathway as a potential OA treatment.

### 3.2. TGF-β Signaling

TGF-β has three isoforms—TGF-β1, TGF-β2, and TGF-β3. These isoforms bind to type I and type II receptors to activate downstream Smad signaling molecules, thereby regulating the transcription of target genes. Among them, Smad2/3 is phosphorylated by the type I receptor ALK5, forms a complex with Smad4, and translocates into the nucleus to regulate the expression of chondrocyte phenotype markers (SOX9, type II collagen, and aggrecan) [55,56]. This pathway promotes chondrocyte differentiation and matrix synthesis, which are crucial for maintaining chondrocyte homeostasis. On the other hand, Smad1/5/8 is phosphorylated by the type I receptor ALK1, forms a complex with Smad4, and translocates into the nucleus to regulate the expression of bone-related genes [57]. Research by van den Bosch et al. demonstrated that the WNT signaling molecule WISP1 can modulate TGF-β signaling by shifting SMAD2/3 phosphorylation mediated by ALK5 toward SMAD1/5/8 phosphorylation mediated by ALK1, thereby inducing a hypertrophic chondrocyte phenotype [58]. An overview of the role of the TGF-β signaling pathway in regulating chondrocyte hypertrophy is provided in Figure 2.

The abnormal activation of the TGF-β signaling pathway may lead to a transition of chondrocytes into a hypertrophic phenotype, resulting in the increased expression of type X collagen, MMP-13, and alkaline phosphatase [57,59,60]. In in vitro chondrogenic differentiation experiments using stem cells, TGF-β1 and TGF-β3 are widely used. Futrega et al. reported that short-term exposure to TGF-β can stably induce chondrogenic differentiation, whereas exposure lasting up to 21 days leads to chondrocyte hypertrophy [61]. Further in vitro and in vivo studies by Pelttari et al. confirmed that under the influence of TGF-β1, chondrocytes undergo hypertrophic differentiation, accompanied by the vascularization and ossification of the cartilage matrix [62]. Clinical reports indicate that TGF-β1 levels are significantly elevated in the joints of OA patients, while almost undetectable in normal joints [63,64]. Moreover, studies in experimental OA mouse models have demonstrated the critical role of the TGF-β signaling pathway in osteophyte formation and chondrocyte hypertrophy [64,65]. These findings provide strong evidence for the association between the TGF-β pathway and chondrocyte hypertrophy in OA.

Given its pivotal role, targeting the TGF-β signaling pathway has emerged as a promising therapeutic strategy for OA. One such approach involves Sprifermin (AS-902330), a recombinant human FGF-18 protein molecule (rhFGF18) and its truncated form. In in vitro studies, this drug has been shown to significantly promote chondrocyte proliferation and ECM production by activating the extracellular-regulated protein kinase (ERK) pathway in chondrocytes. Notably, this effect is maintained in both continuous and intermittent dosing regimens, suggesting its potential therapeutic value in promoting cartilage growth and repair, which may help slow the progression of OA [66]. In clinical research, sprifermin has completed Phase I clinical trials involving 180 knee OA patients. The results show significant improvements in the patients’ condition after the intra-articular injection of the drug. Specifically, the trend of cartilage thickness reduction was alleviated, and cartilage thickness on both the tibia and the joint space was significantly greater than in the placebo group, with a clear dose-dependent effect. Importantly, no severe adverse events were observed during the treatment [67]. Therefore, although sprifermin may become a potential disease-modifying OA drug for OA patients, its therapeutic effect and safety still require more clinical evidence. Another promising agent is Losartan, a TGF-β pathway inhibitor. Studies by Deng et al. revealed that both in vitro and in vivo OA models exhibit PPARγ downregulation concurrent with abnormal TGF-β pathway activation. Treatment with Losartan was shown to upregulate PPARγ expression, suppress the overactive TGF-β pathway, and reduce inflammatory cytokines such as MMP-13, effectively slowing OA progression [62]. Research by Thomas et al. further validated Losartan’s anti-arthritic effects in animal models of temporomandibular and knee OA. Their findings confirmed that Losartan mitigates arthritis progression by inhibiting the TGF-β pathway, providing strong evidence of its therapeutic potential in OA treatment [63].

In summary, targeting the TGF-β signaling pathway through agents like Sprifermin and Losartan represents a promising avenue for OA therapy, with both demonstrating significant preclinical and early clinical efficacy. Further studies are essential to establish their long-term safety and therapeutic value.

### 3.3. BMP Signaling

During normal bone tissue development, bone morphogenetic proteins (BMPs), such as BMP-2 and BMP-4, promote the differentiation of MSCs into chondrocytes and the production of the cartilage matrix, playing a critical role in maintaining bone tissue homeostasis [68,69,70]. However, in the pathological process of OA, the abnormal activation of BMP-2 signaling in pre-hypertrophic and hypertrophic chondrocytes leads to chondrocyte hypertrophy, thereby accelerating cartilage degeneration. In this process, BMP-2 binds to its type I and II receptors, which activates the phosphorylation of downstream signaling molecules SMAD1, SMAD5, and SMAD8. This, in turn, activates the transcription factor RUNX2, inducing the hypertrophic differentiation of chondrocytes and increasing the expression of IHH and type X collagen. Therefore, the canonical BMP Smad1/5/8 signaling pathway is closely associated with chondrocyte hypertrophy and endochondral ossification [71,72]. An overview of the role of the BMP signaling pathway in regulating chondrocyte hypertrophy is provided in Figure 2.

To confirm the correlation between BMP signaling activity and OA, Chawla et al. established a three-dimensional (3D) in vitro OA model using OA chondrocytes. By restricting BMP signaling with BMP type I receptor inhibitors, the results showed a reduction in OA hypertrophic features while maintaining the synthesis of cartilage extracellular matrix [73]. Additionally, to elucidate the relationship between BMP and OA in connection with the Wnt signaling pathway, Papathanasiou et al. demonstrated that BMP stimulation increased the expression levels of hypertrophic markers in chondrocytes, such as type X collagen, MMP-13, and ADAMTS-5. However, the siRNA-mediated knockdown of the Wnt pathway receptor LRP5 abolished these hypertrophic effects, indicating that BMP signaling may promote OA progression by enhancing Wnt/β-catenin signaling [74]. In contrast, BMP-7 has been shown to aid cartilage repair and inhibit cartilage degradation [48,75,76,77]. Caron et al. further demonstrated that the anti-hypertrophic effect of BMP-7 on OA chondrocytes was reflected in the reduced expression levels of type X collagen, MMP-13, and RUNX2. This effect was achieved through the BMP-7-mediated inhibition of the hypertrophic phenotype of OA chondrocytes via BAPX1/NKX3.2 [2].

BMP signaling pathway inhibitors have also been proposed to slow chondrocyte hypertrophy and OA progression. In a rat OA model, Chien et al. administered an intra-articular injection of the BMP-2 inhibitor Noggin. The results show that Noggin reduced the expression of the inflammatory cytokine IL-1β and BMP-2, effectively alleviating cartilage degeneration and the progression of OA [78]. Dorsomorphin selectively inhibits BMP type I receptors, thereby blocking the BMP-mediated phosphorylation of SMAD1/5/8, showing potential in inhibiting chondrocyte hypertrophy and osteogenesis, and opening new pathways for OA treatment [79]. Chawla et al. applied the specific BMP type I receptor inhibitor LDN193189 in both in vitro and in vivo OA models. The results show that the inhibitor significantly downregulated the expression of chondrocyte hypertrophy markers, including collagen X and MMP-13, demonstrating its potential to inhibit chondrocyte hypertrophy [80]. It is well known that BMP-7 inhibits the hypertrophic phenotype of chondrocytes, alleviates cartilage degeneration, and promotes cartilage repair [76,77]. Caron et al. screened two peptide sequences, p [63,64,65,66,67,68,69,70,71,72,73,74,75,76,77,78,79,80,81,82] and p [74,77,78,81,82,83,84,85,86,87,88,89,90,91,92,93,94,95], from the BMP-7 peptide library that can effectively attenuate the OA chondrocyte phenotype. In a rat OA model, they confirmed that these peptide sequences reduce the expressions of chondrocyte hypertrophy markers, including collagenX and MMP-13, by modulating SMAD signaling activity, offering potential novel therapeutic agents for OA treatment [95].

### 3.4. WNT Signaling

The WNT signaling pathway plays an essential role in numerous developmental processes, including bone development. In the canonical WNT signaling pathway, WNT ligands bind to the Frizzled receptor and LRP5/6 co-receptor, inhibiting the activity of the β-catenin degradation complex and thereby increasing β-catenin levels in the cytoplasm. This accumulated β-catenin then translocates into the nucleus, where it binds to TCF/LEF transcription factors, activating downstream gene transcription and further regulating the expression of the critical bone formation factor RUNX2 [96]. An overview of the role of the WNT signaling pathway in regulating chondrocyte hypertrophy is provided in Figure 2. Research has demonstrated that WNT pathway activation promotes the differentiation of chondroprogenitor cells into osteoblasts while inhibiting their differentiation into chondrocytes [97,98]. Additionally, aberrant WNT pathway activation enhances RUNX2 activity, driving the hypertrophic differentiation of chondrocytes, a process closely linked to the progression of OA [99,100].

SM04690, also known as lorecivivint, is a small-molecule inhibitor of the Wnt signaling pathway that has shown potential as a disease-modifying agent in various clinical settings. This compound has been evaluated for its effects on degenerative disc disease (DDD), OA, and temporomandibular joint OA, among other conditions. In the context of degenerative disc disease, SM04690 has demonstrated beneficial effects on intervertebral disc cells in vitro, and has reduced disease progression in a rat model of DDD. The compound was shown to inhibit Wnt pathway gene expression, decrease cell senescence, and promote chondrocyte-like differentiation, suggesting its potential as a therapeutic agent for altering the progression of DDD in humans [101]. For OA, SM04690 has been evaluated in both in vitro and in vivo studies. It has been shown to induce the differentiation of human MSCs into chondrocytes and decrease cartilage catabolic marker levels. In a rodent model of OA, a single intra-articular injection of SM04690 resulted in increased cartilage thickness and protection from cartilage catabolism, indicating its potential as a disease-modifying therapy for OA [102]. Another study showed that EZH2 is the catalytic subunit of Polycomb Repressive Complex 2 (PRC2). Research by Chen et al. has shown that the expression of EZH2 is significantly elevated in chondrocytes from OA patients. The overexpression of EZH2 promotes the high expression of hypertrophic markers such as collagen X, MMP-13, and ADAMTS-5, and this process has been confirmed to be closely related to the activation of the WNT signaling pathway. The EZH2 inhibitor EPZ005687 successfully reversed these changes by silencing WNT signaling, thereby delaying the progression of OA in mice [84]. In another study, Rojas et al. treated MSCs in an in vitro chondrogenic differentiation experiment with the WNT signaling inhibitor Dickkopf-1 (DKK1). After 21 days of observation, they found that this treatment inhibited the terminal differentiation of chondrocytes, downregulated hypertrophic markers such as collagenX and ALP, and upregulated the expression of typeII collagen and proteoglycans. These results suggest that, when the WNT signaling pathway is inhibited, DKK1 can maintain the phenotype of mature chondrocytes and prevent their differentiation into a hypertrophic phenotype, highlighting the potential of DKK1 for use in the treatment of OA [85].

### 3.5. FGF Signaling

The FGF family consists of 23 multifunctional signaling molecules extensively involved in various physiological processes, including cell proliferation, differentiation, angiogenesis, wound healing, embryonic development, and metabolic regulation [80,103,104,105,106]. Moreover, FGF is closely related to the pathological processes of OA [36,107,108]. Under normal conditions, FGF-2 plays a critical role in maintaining cartilage homeostasis by balancing anabolic and catabolic activities. However, in OA chondrocytes, the expression level of FGFR1 is significantly elevated [98]. When FGF-2 binds to FGFR1, it induces receptor phosphorylation, subsequently activating Ras and protein kinase C delta (PKCδ). Ras transmits signals to the Raf-MEK1/2-ERK1/2 cascade, while PKCδ activates the p38 and JNK signaling pathways [109,110]. These three mitogen-activated protein kinase (MAPK) pathways (ERK, p38, and JNK) act synergistically to activate the transcription factors RUNX2 and Elk-1. Elk-1, in particular, further transactivates the expression of MMP-13, thereby accelerating the degradation of the cartilage extracellular matrix [36,54,55,111,112,113]. Additionally, phosphatidylinositol phosphate kinase (PI3K) is another critical signaling pathway involved in the FGF-mediated regulation of osteogenesis. Studies have demonstrated that FGF-2 or FGF-4 activates the PI3K-Akt pathway, significantly promoting the proliferation of osteoblast precursor cells and the differentiation of osteoblasts [114,115]. An overview of the role of the FGF signaling pathway in regulating chondrocyte hypertrophy is provided in Figure 3.

Research by Yan et al. indicated that the expression of FGF-8 and FGFR3 is significantly increased in hypertrophic chondrocytes [116], and stimulation by inflammatory cytokines IL-1β and TNF-α promotes the release of various proteolytic enzymes, contributing to cartilage matrix degradation [78,117]. Another study demonstrated that FGF-23 and FGFR1 are not only highly expressed in hypertrophic chondrocytes of the growth plate [79], but are also more prominently expressed in OA chondrocytes [118]. Treatment with exogenous FGF-23 significantly upregulated RUNX2 expression in both OA and normal chondrocytes [117]. Bianchi et al. found that the exposure of OA chondrocytes to exogenous FGF-23 led to a marked increase in hypertrophic markers, such as typeX collagen, MMP-13, and vascular endothelial growth factor (VEGF) [118]. These findings highlight the critical role of the FGF signaling pathway in chondrocyte hypertrophy and OA progression.

FGF signaling pathway inhibition is another approach to reducing chondrocyte hypertrophy in OA. Xu et al. applied the FGFR1 inhibitor G141 in both in vitro cell models and animal models. The results demonstrated that, in vitro, G141 significantly reduced the expression of proteases in chondrocytes and decreased the phosphorylation levels of MAPK, while notably reducing the loss of proteoglycans. In animal models, G141 effectively protected articular cartilage from arthritis-induced damage and alleviated the progression of OA [86].

### 3.6. Hypoxia Signaling

Compared to tissues with abundant blood flow, articular cartilage exists in a relatively hypoxic environment, as it relies on synovial fluid for nutrients and oxygen; this state is referred to as physiological hypoxia. In such an environment, hypoxia-inducible factors (HIFs), which sense changes in oxygen tension, play an important role in cartilage. HIFs include HIF-1α, HIF-2α, and HIF-3α. Among these, HIF-1α and HIF-2α are closely related to the maintenance and alteration of the chondrocyte phenotype. Studies have shown that HIF-1α enhances the chondrocyte phenotype and inhibits hypertrophic differentiation [37], whereas HIF-2α promotes hypertrophic differentiation by upregulating the expressions of collagen X, MMP-13, and VEGF [37]. Thus, maintaining a balance between HIF-1α and HIF-2α in hypoxic environments is crucial for preserving cartilage homeostasis. In OA, HIF-1α protects articular cartilage by promoting the chondrocyte phenotype and enhancing anabolic activity [95]. In contrast, HIF-2α promotes chondrocyte apoptosis and inflammatory responses, leading to cartilage degradation through catabolic processes [21]. Additionally, HIF-2α not only upregulates the expression of COL10A1, RUNX2, MMP-13, and VEGF during endochondral ossification [108,116], but is also highly expressed in chondrocytes in OA. Therefore, HIF-2α represents a potential therapeutic target for OA. An overview of the role of the hypoxia signaling pathway in regulating chondrocyte hypertrophy is provided in Figure 3.

MicroRNAs targeting HIF-2α have also shown promise in inhibiting chondrocyte hypertrophy, maintaining cartilage homeostasis, and preventing matrix degradation. MicroRNAs such as miR-365, miR-96-5p, and miR-96-3p have been highlighted for their efficacy in this regard [87,88,89].

### 3.7. Inflammatory Signaling

OA is a chronic low-grade inflammatory condition marked by the sustained activation of NF-kB and VEGF signaling pathways triggered by inflammatory cytokines. This prolonged activation promotes chondrocyte hypertrophy, thereby accelerating disease progression. An overview of the role of the inflammatory (NF-κB and VEGF) signaling pathway in regulating chondrocyte hypertrophy is provided in Figure 3.

#### 3.7.1. NF-κB Signaling

The NF-κB signaling pathway is involved in responses to various stimuli, including inflammatory cytokines, free radicals, ultraviolet radiation, and microbial infections. It plays a crucial role in regulating immune and inflammatory responses, as well as processes such as cell growth and apoptosis [119]. In OA, the expression levels of inflammatory cytokines such as IL-1β, IL-6, and TNFα are significantly higher than in healthy individuals, and as the disease progresses, the expressions of these inflammatory mediators further increase [120]. These inflammatory cytokines activate IκB kinase, leading to the phosphorylation of IκB proteins, which are then degraded through ubiquitination, thereby releasing NF-κB from its inhibition. The liberated NF-κB subsequently translocates to the nucleus, where it regulates the transcription of various target genes [121]. In OA, inflammatory stimuli activate the MAPK/RUNX2 signaling axis through inflammatory cytokines IL-1β, IL-6, and TNF-α, inducing chondrocyte hypertrophy and promoting the synthesis and secretion of MMP-13 and ADAMTS-5 [122,123,124]. In this process, the NF-κB signaling pathway acts as a key regulatory factor, playing a central role in promoting chondrocyte hypertrophy and advancing the progression of OA [125,126,127,128,129].

The NF-κB signaling pathway plays a detrimental role in OA, and natural compounds like thymoquinone (TQ) and curcumin have been explored for their potential to inhibit it [90,91]. Wang et al. discovered that TQ inhibited the expressions of multiple inflammatory cytokines in a concentration-dependent manner in an IL-1β-induced OA in vitro model. Furthermore, the study demonstrated that this inhibitory effect was achieved through the regulation of the NF-κB and MAPKs signaling pathways [90]. In another study, Buhrmann et al. tested the anti-inflammatory effects of curcumin using an in vitro model that mimicked the OA microenvironment. The results show that curcumin inhibited the activation of NF-κB, promoted the expression of chondrocyte phenotype markers, and effectively reduced inflammation and chondrocyte apoptosis in the OA microenvironment [91]. Collectively, the ability of these natural compounds to inhibit the NF-κB signaling pathway highlights their potential applicability in the treatment of OA.

#### 3.7.2. VEGF Signaling

The binding of the vascular endothelial growth factor (VEGF) to its receptor activates downstream signaling molecules PKC and PI3K, which promote endothelial cell proliferation through the PKC/MAPK and PI3K/Akt signaling pathways, thereby inducing angiogenesis [130,131]. During endochondral ossification, VEGF induces vascularization at the ossification center. At the stage of chondrocyte hypertrophy, the production of type II collagen and proteoglycans decreases, while the expressions of MMP-13, alkaline phosphatase, VEGF, and RUNX2 increase [132]. Studies have shown that VEGF levels in the serum of OA patients are higher than in normal individuals, and their concentrations rise as the disease progresses [120]. Additionally, VEGF expression is often upregulated in the joint environment in response to mechanical stress and inflammatory reactions, leading to increased cartilage vascularization. This process is observed in OA pathology, where it manifests as chondrocyte hypertrophy followed by cartilage degradation [133]. Mechanical forces applied to joints can induce changes in the expression of VEGF and its receptors, further contributing to the complex pathology of the disease. For instance, cyclic mechanical loading has been shown to enhance VEGF expression in chondrocytes, leading to hypertrophic responses and the overall deterioration of cartilage integrity. This indicates that the VEGF signaling pathway not only promotes angiogenesis, but also influences chondrocyte behavior, resulting in chondrocyte hypertrophy and the loss of cartilage homeostasis [134]. Moreover, angiogenesis is closely linked to inflammation. Inflammatory mediators, mainly cytokines secreted by inflammatory cells such as macrophages, can stimulate angiogenesis, including the secretion of VEGF [135]. Angiogenesis promotes chondrocyte hypertrophy, and hypertrophic chondrocytes further secrete VEGF, intertwining inflammation with angiogenesis. This interplay ultimately disrupts cartilage homeostasis, exacerbating the progression of OA [136,137].

VEGF signaling, which promotes angiogenesis and chondrocyte hypertrophy, has also been targeted. Marsano et al. used retroviral transduction to enable human MSCs to express a decoy soluble VEGF receptor-2 (sFlk1). The receptor effectively sequesters endogenous VEGF, thereby inhibiting angiogenesis. The sFlk1-expressing MSCs were seeded onto collagen sponges and implanted into nude mice to assess their cartilage formation potential. After 12 weeks, the results showed that these cells formed stable hyaline cartilage with no signs of angiogenesis, and there was no expression of type X collagen or MMP-13, indicating no hypertrophic differentiation [130]. The study suggests that blocking VEGF to inhibit hyper-trophic chondrocyte differentiation and promote stable cartilage formation is an effective strategy.

### 3.8. YAP Signaling

The Hippo signaling pathway regulates organ size by controlling cell proliferation and apoptosis, and responds to mechanical signal changes. As a key transcription factor in this pathway, Yes-associated protein (YAP) plays a significant role not only in cartilage development, but also in maintaining cartilage homeostasis [138]. Studies have shown that YAP is both essential and sufficient for maintaining cartilage homeostasis in OA [129,132]. Deng et al. revealed the dual functions of YAP during endochondral ossification; it promotes chondrocyte proliferation in the early stage, while inhibiting chondrocyte hypertrophy and maturation in the later stage [139]. Furthermore, Zhang et al. found that the localization of YAP is closely related to matrix stiffness. On softer matrices, YAP predominantly localizes in the cytoplasm; however, as matrix stiffness increases, YAP gradually translocates from the cytoplasm to the nucleus [140,141]. During the early stages of cartilage development, high levels of phosphorylated YAP retain it in the cytoplasm, thereby maintaining the chondrocyte phenotype. Conversely, in the later stages of chondrocyte differentiation, the high expression of nuclear YAP is closely associated with chondrocyte hypertrophy [104,105,135]. Using biomimetic materials to simulate cartilage matrix environments, Lee et al. demonstrated that softer matrices promote chondrocyte proliferation, whereas stiffer matrices induce chondrocyte hypertrophic differentiation [142]. These findings suggest that YAP regulates cartilage homeostasis through a biphasic mechanism. Further research revealed that YAP expression is significantly upregulated in OA chondrocytes. Gong et al. found that YAP overexpression exacerbates the expression of hypertrophic markers such as MMP-13 and ADAMTS-5 induced by inflammatory factors like IL-1β, accelerating cartilage matrix degradation and driving OA progression [143]. These findings underscore the critical role of YAP in regulating cartilage homeostasis and the pathogenesis of OA. An overview of the role of the YAP signaling pathway in regulating chondrocyte hypertrophy is provided in Figure 3.

Zhang et al. utilized PDMS substrates with varying stiffnesses to simulate the matrix stiffness of cartilage under both physiological and pathological conditions, investigating how different matrix stiffnesses affect YAP activity in chondrocytes. The results reveal that YAP was activated on stiffer substrates, leading to the loss of the chondrocyte phenotype. However, in YAP gene knockout or YAP inhibitor experiments, chondrocytes were protected, further confirming that YAP activation accelerates cartilage degradation. These findings suggest that YAP could be a potential therapeutic target for OA [144]. In another study, Lee et al. synthesized hydrogels using eight-arm polyethylene glycol acrylates (PEG) and oxidized methacrylated alginate to replicate the cartilage matrix microenvironment and encapsulated MSCs to observe changes in chondrogenic differentiation. The results indicate that, in substrates with rapid degradation and soft stiffness, the chondrocyte phenotype was well maintained without hypertrophy, whereas on substrates with slow degradation and higher stiffness, chondrocytes exhibited hypertrophy [142]. Therefore, by modulating the stiffness of biomaterials used to mimic the cartilage matrix, this approach may provide an effective strategy for inhibiting chondrocyte hypertrophy and slowing the progression of arthritis.

### 3.9. mTOR Signaling

The mammalian target of rapamycin (mTOR) is a nutrient-sensing protein kinase that responds to signals from nutrients and growth factors via the Akt/PI3K signaling pathway, regulating various physiological processes such as cell growth and metabolism [112,145,146]. Research by Yan et al. demonstrated that mTOR signaling promotes chondrocyte proliferation during the early stages by regulating the transcription of PTHrP, while inhibiting terminal differentiation in later stages, making it a critical coordinator in the process of chondrocyte hypertrophic differentiation [147]. Phornphutkul et al., using the ATDC5 chondrocyte cell line, found that mTOR signaling promotes chondrocyte differentiation by regulating the expression of IHH [148]. Additionally, Zhang et al. activated mTOR signaling in the articular cartilage of mice, and observed an upregulation of IHH expression, along with increased levels of hypertrophic markers RUNX2 and type X collagen, accompanied by symptoms of OA [149]. Interleukin-18 (IL-18) is a pro-inflammatory cytokine secreted by chondrocytes and macrophages, closely associated with the development of OA [150]. Bao et al. revealed that IL-18 induces an inflammatory state in rat chondrocytes, and the activation of the PI3K/Akt/mTOR signaling pathway is the primary cause of the downregulation of the chondrocyte phenotype [151]. These findings suggest that the PI3K/Akt/mTOR signaling pathway not only plays a crucial role in physiological endochondral ossification, but also mediates chondrocyte hypertrophy and OA through mTOR activation, further highlighting its key regulatory role in chondrocyte proliferation and differentiation [152]. An overview of the role of the mTOR signaling pathway in regulating chondrocyte hypertrophy is provided in Figure 3.

Finally, mTOR signaling inhibitors like rapamycin have been explored for their therapeutic effects. Bao et al. successfully reversed the progression of OA in an IL-18-induced rat model by using the mTOR-specific inhibitor rapamycin [151]. Zhou et al. provided valuable insights into OA treatment from the perspective of microRNAs that play a crucial role in post-transcriptional regulation. They first confirmed the abnormal expression of the circular RNA ciRS-7 (ciRS-7)/microRNA 7 (miR-7) axis, which is associated with the mTOR signaling pathway in OA. Further studies found that miR-7-siRNA could alleviate cartilage damage in mechanically induced OA rats, while the overexpression of miR-7 exacerbated this damage [152]. In summary, the mTOR signaling pathway plays an important role in OA-related chondrocyte differentiation. Strategies targeting the inhibition or modulation of this pathway could provide innovative therapeutic interventions for OA.

## 4. Tissue Engineering Approaches for Suppressing Chondrocyte Hypertrophy

Cartilage tissue engineering is an interdisciplinary science that aims to repair damaged cartilage and restore its function by combining engineering principles, chondrocyte seed cells, biomaterial scaffolds, and bioactive factors that promote chondrocyte growth and maintain their phenotype stability [93,94]. However, during the manipulation of seed cells, the issue of chondrocyte hypertrophy remains a significant challenge that needs to be addressed. The following discussion will explore strategies to counteract chondrocyte hypertrophy from three aspects—seed cell implantation, biomaterials, and growth-active factors.

### 4.1. Cells

In cartilage tissue engineering research, common cell sources include embryonic stem cells (ESCs), induced pluripotent stem cells (iPSCs), chondroprogenitor cells, and MSCs derived from various tissues. However, the use of ESCs and iPSCs is heavily restricted in the field of tissue engineering due to ethical concerns and the risk of teratomas. Therefore, this chapter will focus on chondroprogenitor cells and MSCs, discussing strategies to prevent hypertrophy during chondrocyte differentiation.

In 1994, the New England Journal of Medicine reported the first case of treatment using autologous chondrocyte implantation (ACI) [153]. Studies have shown that ACI significantly alleviates pain in patients, with encouraging clinical outcomes. As a result, ACI has been regarded as the “gold standard” and preferred method for the biological repair of cartilage defects [154,155]. Since then, many companies specializing in the in vitro expansion of autologous chondrocytes have rapidly emerged. However, ACI has a critical limitation: during the in vitro expansion of chondrocytes, dedifferentiation occurs, leading to the loss of their phenotype and a decline in mechanical properties, which becomes a major obstacle to the successful repair of cartilage defects.

Typically, the in vitro expansion of chondrocytes is initially conducted on tissue culture plastic surfaces. When cells are cultured in two-dimensional planes, the expression of cartilage markers, such as type II collagen and proteoglycans, gradually decreases with increasing passage numbers, and can even be completely lost. Meanwhile, the expression of hypertrophic chondrocyte markers, such as type I collagen, type X collagen, and ALP, increases [156,157,158]. To overcome the dedifferentiation of chondrocytes, high-density culture is considered an effective strategy. Schulze-Tanzil et al. encapsulated dedifferentiated chondrocytes from the fourth passage and those from passages 5–8 in alginate beads for high-density culture. They found that fourth-passage cells regained the chondrocyte phenotype, whereas cells from passages 5–8 did not exhibit such redifferentiation [159]. Another approach to promoting the restoration of the chondrocyte phenotype in dedifferentiated chondrocytes is the application of growth factors. Insulin-like growth factor I (IGF-I) and transforming growth factor β (TGF-β) have been shown to induce monolayer expanded chondrocytes to regain their chondrogenic differentiation potential [160,161,162].

MSCs were first isolated from bone marrow by Pittenger et al. and demonstrated to possess multipotent differentiation potential [163]. Subsequently, researchers discovered the presence of MSCs in various other tissues, including adipose tissue, muscle, dermis, articular cartilage, and even perinatal tissues [164,165].

MSCs, due to their unique biological properties, are widely regarded as strong candidates for cartilage regeneration, supported by the differentiation and paracrine theories [166]. The differentiation theory suggests that MSCs can directly differentiate into chondrocytes, thereby replacing damaged cartilage tissue [94]. Meanwhile, the paracrine theory highlights that MSCs secrete various bioactive factors to modulate the local microenvironment, creating favorable conditions for cartilage regeneration [167]. Although MSC-derived chondrocytes can maintain their phenotype in the short term and partially alleviate pathological changes caused by cartilage damage, they often undergo hypertrophy over time under the influence of the surrounding microenvironment. This hypertrophy can lead to further differentiation into osteoblasts, ultimately resulting in the replacement of damaged cartilage with fibrocartilage [168,169]. Therefore, controlling the hypertrophic differentiation of MSC-derived chondrocytes is a critical strategy for effective cartilage repair. Multiple signaling pathways are involved in regulating the hypertrophic phenotype during the differentiation of MSCs into chondrocytes, including WNT, BMP, TGF-β, PTHrP/IHH, FGF, IGF, and HIF pathways [170,171]. In the WNT signaling pathway, inhibitors such as Dickkopf (DKK1) and Frizzled-related protein (FRZB) can upregulate the expression of type II collagen and proteoglycans. Unfortunately, they have no significant effect on the expression of type X collagen. In the BMP signaling pathway, BMP-7 and its inhibitor Gremlin 1 (GREM1) effectively prevent chondrocyte hypertrophy and reduce ossification [172,173]. While TGF-β is a potent growth factor for in vitro chondrogenesis, studies by Chen et al. have revealed another aspect of TGF-β—it may induce hypertrophic changes in chondrocytes and promote the expression of angiogenic factors. Therefore, the potential adverse effects of TGF-β must be carefully considered when using it for cartilage repair [174]. The PTHrP/IHH signaling pathway plays a vital role in endochondral ossification. IHH regulates the expression of PTHrP to coordinate the ossification process. IHH promotes chondrocyte hypertrophy and mineralization, whereas PTHrP inhibits hypertrophic differentiation through negative feedback on IHH [175]. Co-culture experiments involving chondrocytes and MSCs have further demonstrated that adding PTHrP significantly reduces hypertrophy in both MSCs and differentiated chondrocytes [54,176]. To obtain a large number of MSCs, FGF-2 is commonly used to enhance cell proliferation. However, FGF-2 also upregulates the expression of RUNX2 and type X collagen [177,178], necessitating caution in its use. IGF-1 is a frequently used growth factor in chondrocyte differentiation experiments due to its ability to promote differentiation. However, some studies have shown that IGF-1 can also promote the formation of hypertrophic chondrocytes [179,180]. Therefore, the benefits and drawbacks of IGF-1 should be carefully evaluated based on the research objectives.

### 4.2. Bioactive Factors

Some bioactive factors, such as TGF-β, BMP, FGF, IGF-1, and PTHrP, have been discussed in detail in the cellular section. In the following, we will focus on inhibitors of the cartilage hypertrophy marker MMP-13 and natural bioactive compounds.

Among cartilage hypertrophy markers, MMP-13 is a key protease that degrades the cartilage matrix, and its inhibition is considered an effective strategy to reduce joint cartilage loss. CL82198 is a chemical inhibitor specifically targeting MMP-13. In a surgical injury-induced OA mouse model, Wang et al. demonstrated that daily injections of CL82198 for 16 weeks significantly slowed the progression of OA, maintained normal levels of type II collagen and proteoglycans, and effectively inhibited chondrocyte apoptosis. Furthermore, the study revealed that CL82198 could block more than 90% of MMP-13 activity [181]. These findings suggest that using CL82198 to inhibit MMP-13 is a promising strategy for mitigating the progression of OA.

Traditional herbal medicines have attracted significant attention for their remarkable therapeutic effects and relatively lower toxicity compared to synthetic compounds, particularly in the prevention and treatment of chondrocyte hypertrophy and OA. Cao et al. investigated the effects of curcumin on the chondrogenic differentiation and hypertrophic changes of MSCs. The results show that while curcumin did not significantly impact the cartilage phenotype, it effectively inhibited chondrocyte hypertrophy by regulating the IHH and Notch signaling pathways [182]. In another study, Huang et al. confirmed the anti-inflammatory effects of vanillic acid using both in vitro and in vivo OA models. The results demonstrate that vanillic acid inhibited the production of inflammatory cytokines and cartilage matrix-degrading enzymes, reduced the expression of hypertrophic chondrocyte markers such as type X collagen, RUNX2, and vascular endothelial growth factor (VEGF), and alleviated the progression of OA by inhibiting the activation of the MAPK and PI3K/AKT/NF-κB signaling pathways [183]. Luo et al. also demonstrated, through in vitro cell models of OA and joint injury animal models, that icariin can increase the expression of SOX9, type II collagen, and proteoglycans, while upregulating the expression of parathyroid hormone-related protein. Additionally, it reduced the levels of hypertrophic chondrocyte markers, including type X collagen and MMP-13, and downregulated the expression of IHH, thereby slowing the progression of OA [184]. To address the issue of mesenchymal stem cell-derived chondrocyte hypertrophy, Cao et al. conducted a study on cordycepin. The results reveal that cordycepin promoted the expression of chondrocyte phenotype markers SOX9 and type II collagen by inhibiting Nrf2 and activating the BMP signaling pathway. At the same time, it suppressed the expression of chondrocyte hypertrophy markers, including type X collagen and RUNX2, through the PI3K/Bapx1 and Notch signaling pathways [185].

### 4.3. Biomaterials

Researchers have utilized biomaterials to replicate the three-dimensional microenvironment of tissues, achieving not only the inhibition of chondrocyte hypertrophy but also the effective restoration of the chondrocyte phenotype [186]. Biomaterials used in tissue engineering are generally categorized into natural and synthetic types. Among them, natural biomaterials—such as collagen, silk fibroin, alginate, chitosan, and cartilage matrix—have demonstrated significant potential in suppressing chondrocyte hypertrophy.

In studies involving collagen hydrogels for chondrocyte culture, the results have revealed that these hydrogels promote the expression of type II collagen and proteoglycans while reducing the expression of type I and type X collagen. This confirms that collagen hydrogels help maintain the chondrocyte phenotype and inhibit hypertrophic differentiation [187,188]. Similarly, Bhardwaj et al. demonstrated in their experiments with silk fibroin 3D scaffolds for the chondrogenic differentiation of adipose-derived MSCs that silk fibroin enhances the expression of SOX9, type II collagen, and proteoglycans, while downregulating markers of chondrocyte hypertrophy, such as type X collagen and MMP-13. They attributed these results to the favorable mechanical and biological microenvironment provided by the silk fibroin scaffolds for cartilage formation [189]. Schulze-Tanzil et al. used alginate hydrogels for the high-density culturing of dedifferentiated chondrocytes, and found that these chondrocytes underwent redifferentiation, successfully restoring the chondrocyte phenotype while reducing the expression of hypertrophy markers [159]. Chitosan, a widely used biomaterial in cartilage tissue engineering, has shown remarkable value in various studies. It promotes the differentiation of MSCs into chondrocytes, inhibits the expression of type X and type I collagen, and reduces the occurrence of chondrocyte hypertrophy [190,191,192]. Rac1, a small GTPase, serves as a crucial positive regulator of chondrogenesis and chondrocyte hypertrophy [193,194]. Research by Zhu et al. revealed that IL-1β enhances Rac1 activity, and activated Rac1 promotes the expression of MMP-13, ADAMTS-5, RUNX2, and type X collagen in chondrocytes through the β-catenin signaling pathway. In a mouse model of OA established by anterior cruciate ligament transection, the intra-articular injection of chitosan microspheres loaded with the Rac1 inhibitor NSC23766 effectively slowed the progression of OA [195].

Cartilage matrix components, such as hyaluronic acid, chondroitin sulfate, and decellularized matrix, play a distinctive role in preventing chondrocyte hypertrophy and delaying the onset of OA. Specifically, chondroitin sulfate, when incorporated into PEG-based hydrogels, enhances the differentiation of MSCs into chondrocytes and inhibits the formation of hypertrophic cartilage [196]. Dynamic loading has been found to further amplify the inhibitory effect of chondroitin sulfate on chondrocyte hypertrophy [197]. Hyaluronic acid hydrogels provide an excellent microenvironment for cartilage formation. When combined with collagen and chondroitin sulfate, they synergistically promote chondrogenic differentiation MSCs and inhibit chondrocyte hypertrophy [198]. However, the crosslinking density of hyaluronic acid significantly influences chondrocyte hypertrophy and matrix calcification, with higher densities promoting these undesirable effects. Therefore, the careful consideration of crosslinking density is essential during the preparation of hyaluronic acid hydrogels [199]. The decellularized matrix, prepared by removing cells and antigenic components while preserving the integrity of other cartilage matrix components, has also shown promise. Chang et al. cultured synovium-derived MSCs cells in a collagen hydrogel containing decellularized cartilage matrix, and observed the enhanced expression of type II collagen and the inhibition of chondrocyte hypertrophy [200]. These findings highlight the ability of cartilage matrix components to create a microenvironment conducive to cartilage growth and differentiation, effectively maintaining the chondrocyte phenotype when incorporated into 3D biomaterial structures.

### 4.4. 3D Bioprinting for Cartilage Injury Repair

Three-dimensional bioprinting is regarded as one of the most promising technologies in cartilage tissue engineering, overcoming many long-standing challenges in the field. It offers new opportunities, particularly in repairing articular cartilage defects and degeneration. The biological scaffolds generated through 3D bioprinting can serve as ECM for cartilage, promoting the formation of new cartilage. Bioinks, the critical components in constructing 3D bioprinted scaffolds, are generally composed of three main elements, as follows: biomaterials (e.g., alginate, collagen, and hyaluronic acid), cells (e.g., chondroprogenitor cells or MSCs), and bioactive molecules (e.g., growth factors, cytokines, or small molecules), providing structural support for cartilage formation and regeneration [201,202,203,204]. Cui et al. utilized thermal inkjet technology for bioprinting, employing bioinks composed of poly (ethylene glycol)dimethacrylate loaded with human chondrocytes, supplemented with FGF-2 and/or TGF-β1. The study demonstrated that the combination of FGF-2 and TGF-β1 significantly enhanced cell proliferation and chondrocyte phenotype compared to the TGF-β1-only group, leading to notable improvements in osteochondral defect repair [205,206].

However, due to the irregular shapes and variable depths of cartilage defects caused by OA, conventional bioprinting techniques face significant challenges during treatment. To address this issue, researchers proposed in situ 3D bioprinting technology. This approach involves scanning the defect area and directly printing the patient’s cells, biomaterials, and bioactive molecules onto the damaged site to create implants that precisely match the target defect [207,208]. Gatenholm et al. successfully implemented precise cell seeding in tibial plateau defects using 3D bioprinting technology. They first scanned an osteoarthritic tibial plateau with cartilage defects retrieved during total knee arthroplasty using a 3D scanner. Bioinks containing human chondrocytes were then used for in situ printing on the defect site. After two weeks of cultivation, significant chondrocyte differentiation was observed, along with high levels of collagen type II and proteoglycan expression. The average cell viability in the printed constructs was 81%. The study concluded that patient-specific in situ bioprinting provides strong evidence for early cartilage tissue regeneration treatments for OA [209]. Additionally, Duchi et al. developed a core–shell GelMA/HAMA scaffold with a stiffness of 200 kPa by exposing GelMA/HAMA hydrogels to 365 nm UV light with an intensity of 700 mW/cm^2^ for 10 s. Adipose-derived stem cells embedded in the scaffold maintained over 90% viability and exhibited an excellent proliferative capacity. In a sheep full-thickness cartilage defect model, they used a handheld in situ bioprinting device for treatment. Eight weeks post-operation, the repaired cartilage showed significantly improved macroscopic and microscopic scores, increased new cartilage formation, and higher collagen type II expression compared to the control group [210,211].

## 5. Conclusions and Perspective

OA is a major chronic disease affecting quality of life in the elderly. To prevent and treat this disease, scientists have conducted extensive research and clinical trials. However, the exact pathogenic mechanism of OA remains unclear, and effective treatment options have yet to be established. The root cause of this issue lies in the insufficient understanding of chondrocytes in both physiological and pathological conditions. Chondrocyte hypertrophy not only occurs during osteogenesis, but also plays a critical role in the onset and progression of OA. YAP, a mechanosensitive factor in the Hippo signaling pathway, is highly expressed in OA, leading to chondrocyte hypertrophy and the subsequently increased synthesis and secretion of MMP-13, severely disrupting the cartilage matrix. CRISPR/Cas9, an advanced genome-editing technology, can be used to knock down the YAP gene in MSCs and chondrocytes, thus reversing chondrocyte hypertrophy [140,212]. Furthermore, knocking down the MMP-13 gene can prevent the upregulation of MMP-13 caused by chondrocyte hypertrophy, offering significant potential for cartilage repair [213,214].

Early observations of morphological and volumetric changes before cartilage degradation are crucial for preventing chondrocyte hypertrophy and the development of OA. However, existing imaging techniques struggle to capture these changes. To detect detailed changes in chondrocyte morphology and volume, confocal laser scanning microscopy [24] and two-photon laser scanning microscopy [25] have been used to specifically label the cytoplasmic space and components of live, in situ chondrocytes in their unaltered ECM with fluorescence. These methods now allow the generation of high-resolution three-dimensional images. Coupled with imaging software, this enables the study and observation of chondrocyte morphology and volume, especially in non-degenerate and degenerative OA chondrocytes, where changes in cell volume and cytoplasmic protrusions can be used for identification and assessment [215,216,217,218,219].

In conclusion, normal chondrocytes are quiescent cells with a low proliferative capacity. In the low-grade inflammatory environment of OA, chondrocyte proliferation is activated, ultimately leading to chondrocyte hypertrophy and the loss of their inherent differentiation phenotype. Although it is not yet clear whether chondrocyte hypertrophy is a trigger for OA, inhibiting hypertrophy is crucial for maintaining normal cartilage function. Chondrocyte hypertrophy is regulated by multiple signaling pathways, so reversing this process through small molecules or tissue engineering approaches requires shifting from single-pathway interventions to multi-pathway strategies. This multi-faceted approach represents an effective strategy to halt or slow the progression of OA.

## Figures and Tables

**Figure 1 bioengineering-12-00077-f001:**
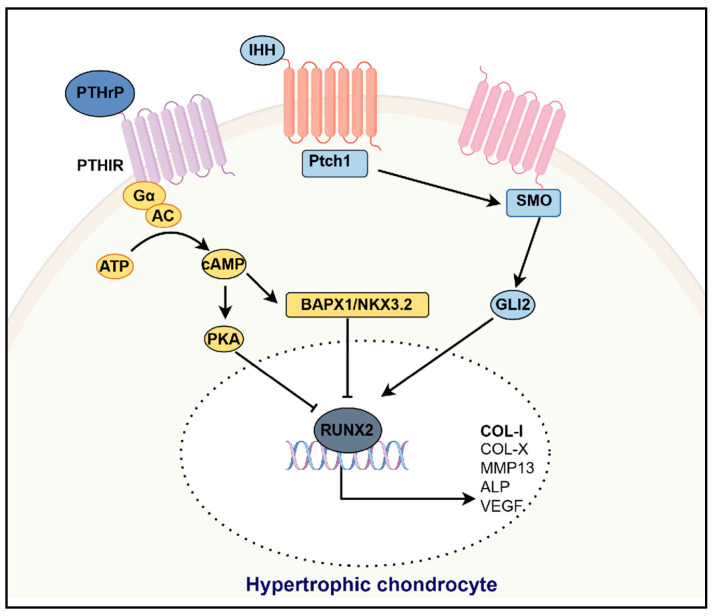
Overview of the role of the IHH/PTHrP signaling pathway in regulating chondrocyte hypertrophy. “→” and “T” represent positive and negative actions, respectively. This figure was created with Figdraw (HOME for Researchers).

**Figure 2 bioengineering-12-00077-f002:**
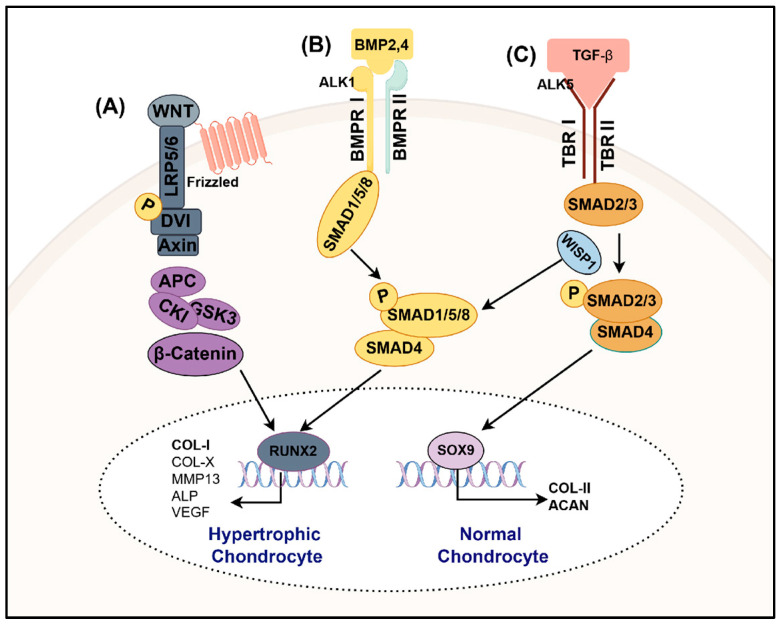
Overview of the roles of the WNT, BMP, and TGF-β signaling pathways in regulating chondrocyte hypertrophy. (**A**) WNT signaling, (**B**) BMP signaling, (**C**) TGF-β signaling. “→” represents positive actions. This figure was created with Figdraw (HOME for Researchers).

**Figure 3 bioengineering-12-00077-f003:**
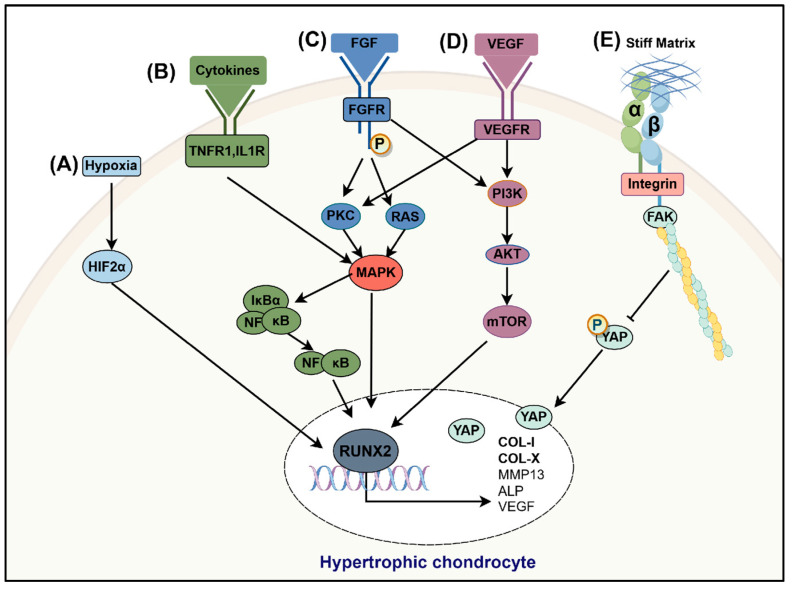
Overview of the roles of the hypoxia, inflammatory (NF-κB and VEGF), FGF, YAP, and mTOR signaling pathways in regulating chondrocyte hypertrophy. (**A**) Hypoxia signaling, (**B**) NF-κB signaling, (**C**) FGF signaling, (**D**) VEGF signaling, (**E**) YAP signaling. “→” and “T” represent positive and negative actions, respectively. This figure was created with Figdraw (HOME for Researchers).

## Data Availability

The data produced in this study are included within the paper.

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
