# Peer review of "Targeting Chondrocyte Hypertrophy as Strategies for the Treatment of Osteoarthritis"

_bioengineering, 2025, doi:10.3390/bioengineering12010077_

Round 1
Reviewer 1 Report
Comments and Suggestions for Authors
1. Figures lack clear legends and sufficient annotations, which reduces their educational value. For instance, Figure 1 summarizing signaling pathways could benefit from an expanded legend explaining arrows, T-shapes, and interactions between pathways.
2. While the article provides a comprehensive overview, it primarily reiterates existing findings without offering novel hypotheses or interpretations that advance the field.
3. The therapeutic strategies are well-discussed but lack specificity in translation to clinical practice. For example, the section on small molecule inhibitors could detail their pharmacokinetics, potential side effects, or clinical trial progress.
4. The discussion on tissue engineering and biomaterials is fragmented. A synthesized framework linking scaffold properties to anti-hypertrophic outcomes would enhance coherence.
5. The repetition of content, such as the role of specific signaling pathways, across different sections detracts from readability. A unified discussion integrating these aspects would improve flow.
6. Cutting-edge approaches such as CRISPR or advanced imaging techniques to study chondrocyte hypertrophy are not addressed. Including these would reflect a more forward-looking perspective.
Author Response
- Figures lack clear legends and sufficient annotations, which reduces their educational value. For instance, Figure 1 summarizing signaling pathways could benefit from an expanded legend explaining arrows, T-shapes, and interactions between pathways.
A: We fully agree with the reviewers' suggestions. To provide a clearer presentation of each signaling pathway, we have divided the original Figure 1 into Figures 1 through 3 and expanded specific pathways to improve clarity and precision.
- While the article provides a comprehensive overview, it primarily reiterates existing findings without offering novel hypotheses or interpretations that advance the field.
A: We fully agree with the reviewer's comments. OA, as a longstanding research topic, has seen little breakthrough in therapeutic strategies to date. Against this backdrop, we conducted an extensive review of relevant literature and proposed "chondrocyte hypertrophy" as a potential breakthrough point for osteoarthritis treatment. Furthermore, we systematically analyzed existing research findings. In addition, we proposed a novel hypothesis suggesting that regulating the YAP signaling pathway could play a critical role in treatment. In our proposed strategies, we elaborated on how to utilize biomaterial matrices to modulate the YAP pathway, thereby inhibiting chondrocyte hypertrophy and halting the progression of OA.
- The therapeutic strategies are well-discussed but lack specificity in translation to clinical practice. For example, the section on small molecule inhibitors could detail their pharmacokinetics, potential side effects, or clinical trial progress.
A: We fully agree with the reviewers' comments. To further explore this issue, we have reviewed a large body of relevant literature, including review articles. Pharmacokinetic studies on small-molecule inhibitors can verify their bioavailability and distribution characteristics in vivo, ensuring that they can effectively reach target tissues and exert their effects. At the same time, preclinical experiments using animal models assess the toxic side effects of small molecules, aiming to ensure efficacy while maximizing safety. As the reviewers pointed out, clinical translational research on small-molecule therapies for osteoarthritis is indeed significantly lacking. This may be due to the fact that the signaling pathways inhibiting chondrocyte hypertrophy are often closely related to other signaling pathways. Although these interventions show significant effects in suppressing chondrocyte hypertrophy, they may interfere with other pathways, leading to unintended adverse effects. Therefore, most studies in the current literature remain at the basic research stage and have yet to achieve clinical translation. In our article, we discussed several small-molecule drug studies related to clinical translation. Based on the reviewers' suggestions, we have added a more detailed and in-depth discussion on this topic, and the supplementary discussion has been highlighted in blue font.
- The discussion on tissue engineering and biomaterials is fragmented. A synthesized framework linking scaffold properties to anti-hypertrophic outcomes would enhance coherence.
A: We agree with the reviewers' comments. The ideal tissue engineering approach is to comprehensively utilize cells, bioactive factors, and biomaterial scaffolds to construct artificial tissues in vitro, which are then transplanted into the body for complete tissue repair and regeneration. However, in the practical application of tissue engineering and regenerative medicine, the choice of single factors, combinations of two factors, or all three elements is often made flexibly based on specific clinical needs and onsite conditions to address practical issues. Based on the reviewers' suggestion, we have applied all three elements in an example of cartilage regeneration, using 3D bioprinting, a cutting-edge technology in tissue engineering, as an example. We have further supplemented the discussion in the "3D Bioprinting for Cartilage Injury Repair" section, and the added discussion has been highlighted in blue font.
- The repetition of content, such as the role of specific signaling pathways, across different sections detracts from readability. A unified discussion integrating these aspects would improve flow.
A: We appreciate the reviewers' valuable suggestions for improving the readability of the article. We have thoroughly reviewed the relevant content regarding the role of specific signaling pathways throughout the manuscript and consolidated the repetitive sections. This adjustment not only avoids redundant descriptions and reduces superfluous content, but also significantly enhances the flow of the article, allowing readers to gain a clearer and deeper understanding of the core content.
- Cutting-edge approaches such as CRISPR or advanced imaging techniques to study chondrocyte hypertrophy are not addressed. Including these would reflect a more forward-looking perspective.
A: We appreciate the reviewer's suggestions. Regarding the cutting-edge methods not covered, such as CRISPR technology or advanced imaging techniques for studying chondrocyte hypertrophy, we have added supplementary discussion in the "Conclusion and Perspective" section and highlighted the additions in blue font.
Reviewer 2 Report
Comments and Suggestions for Authors
In the manuscript entitled “Targeting chondrocyte hypertrophy as strategies for the treatment of osteoarthritis”, authors discussed the underlying mechanisms of chondrocyte hypertrophy in OA, including relationship between osteoarthritis and chondrocyte hypertrophy, potential signaling pathway, and strategies for suppressing chondrocyte hypertrophy. The subject is meaningful but there are some issues to be solved for further publication.
1. The precise pathogenic mechanism underlying osteoarthritis (OA) remains incompletely understood. While the authors aimed to emphasize the role of chondrocyte hypertrophy in the progression of OA in this manuscript, the discussion presented in Section 2 is not sufficiently comprehensive. Moreover, the evidence supporting the therapeutic potential of targeting chondrocyte hypertrophy in OA treatment lacks robustness.
2. In Figure 1, the depiction of signaling pathways involved in regulating chondrocyte hypertrophy is unclear and difficult to interpret, making it challenging for readers to extract meaningful information. The authors are encouraged to reevaluate and revise this figure for clarity and utility.
3. Section 3 identifies several signaling pathways associated with chondrocyte hypertrophy. However, critical pathways such as IHH/PTHrP signaling, TGF-β signaling, BMP signaling, and FGF signaling, which have been extensively studied, are not discussed in sufficient depth. The authors should provide a more comprehensive analysis and incorporate recent findings related to these pathways.
4. In Section 4, the strategies for suppressing chondrocyte hypertrophy are discussed, however, the section's structure is disorganized. For instance, in subsection 4.2.1, the discussion focuses on cellular regulation, which overlaps significantly with the signaling pathways described in Section 3. The rationale for highlighting MMP-13 in this section, while omitting other key factors such as TGF-β, BMP, FGF, IGF-1, and PTHrP, is not adequately justified. Additionally, the content from lines 542 to 550 is redundant, as no specific factors are explicitly addressed. Furthermore, in subsection 4.2.3, the discussion on biomaterials for in vitro differentiation of mesenchymal stem cells (MSCs) or maintenance of the chondrocyte phenotype lacks direct relevance to OA treatment. The inclusion of in vivo evidence is necessary to substantiate the claim that biomaterials can suppress chondrocyte hypertrophy as a therapeutic strategy for OA.
5. The authors are strongly advised to conduct a thorough review of the manuscript to address grammatical and formatting issues.
Author Response
In the manuscript entitled “Targeting chondrocyte hypertrophy as strategies for the treatment of osteoarthritis”, authors discussed the underlying mechanisms of chondrocyte hypertrophy in OA, including relationship between osteoarthritis and chondrocyte hypertrophy, potential signaling pathway, and strategies for suppressing chondrocyte hypertrophy. The subject is meaningful but there are some issues to be solved for further publication.
- The precise pathogenic mechanism underlying osteoarthritis (OA) remains incompletely understood. While the authors aimed to emphasize the role of chondrocyte hypertrophy in the progression of OA in this manuscript, the discussion presented in Section 2 is not sufficiently comprehensive. Moreover, the evidence supporting the therapeutic potential of targeting chondrocyte hypertrophy in OA treatment lacks robustness.
A: We appreciate the reviewers' valuable feedback on our work. In response to the reviewers' concern about the lack of comprehensiveness in the discussion of Section 2, we have expanded the discussion on "The evidence for chondrocyte hypertrophy as a therapeutic target for osteoarthritis" and highlighted the additions in red font.
- In Figure 1, the depiction of signaling pathways involved in regulating chondrocyte hypertrophy is unclear and difficult to interpret, making it challenging for readers to extract meaningful information. The authors are encouraged to reevaluate and revise this figure for clarity and utility.
A: We appreciate the reviewers' valuable feedback on Figure 1. In response to the reviewers' concern that the depiction of signaling pathways regulating chondrocyte hypertrophy was unclear and difficult to interpret, we have split the original Figure 1 into Figures 1 to 3. Additionally, we have expanded some of the signaling pathways to improve the clarity and readability of the figure.
- Section 3 identifies several signaling pathways associated with chondrocyte hypertrophy. However, critical pathways such as IHH/PTHrP signaling, TGF-β signaling, BMP signaling, and FGF signaling, which have been extensively studied, are not discussed in sufficient depth. The authors should provide a more comprehensive analysis and incorporate recent findings related to these pathways.
A: We appreciate the reviewers' valuable feedback on our work. In response to the reviewers' concern that the IHH/PTHrP signaling pathway, TGF-β signaling pathway, BMP signaling pathway, and FGF signaling pathway were not sufficiently discussed in Section 3, we have expanded this section and provided a more comprehensive analysis.
- In Section 4, the strategies for suppressing chondrocyte hypertrophy are discussed, however, the section's structure is disorganized. For instance, in subsection 4.2.1, the discussion focuses on cellular regulation, which overlaps significantly with the signaling pathways described in Section 3. The rationale for highlighting MMP-13 in this section, while omitting other key factors such as TGF-β, BMP, FGF, IGF-1, and PTHrP, is not adequately justified. Additionally, the content from lines 542 to 550 is redundant, as no specific factors are explicitly addressed. Furthermore, in subsection 4.2.3, the discussion on biomaterials for in vitro differentiation of mesenchymal stem cells (MSCs) or maintenance of the chondrocyte phenotype lacks direct relevance to OA treatment. The inclusion of in vivo evidence is necessary to substantiate the claim that biomaterials can suppress chondrocyte hypertrophy as a therapeutic strategy for OA.
A: We appreciate the reviewer’s insightful feedback. In response to the comment regarding the structural confusion in Section 4, which discusses strategies for inhibiting chondrocyte hypertrophy, we have reorganized the content. We incorporated the strategies for each signaling pathway into the latter part of the corresponding signaling pathway section, thereby addressing the structural issues. Additionally, we have presented tissue engineering strategies as a separate section to enhance the clarity and logical flow of the manuscript.Regarding the comment on the overlap between Section 4.1 (cellular regulation) and Section 3 (signaling pathways), we would like to clarify that Section 3 focuses on how various signaling pathways induce chondrocyte hypertrophy, leading to the onset of OA. In contrast, Section 4.1 emphasizes how these pathways can be leveraged to prevent hypertrophy in stem cell-derived chondrocytes, providing valuable guidance for preparing high-quality cell sources for OA treatment. Thus, while there may appear to be overlap, it is indeed a crucial aspect of the manuscript.In response to the issue of "the content from lines 542 to 550 being somewhat redundant as it does not specify specific factors," we have deleted this section.In response to the concern about the emphasis on MMP-13, we highlight its importance because it is a direct pathogenic factor responsible for cartilage damage in OA. While other factors such as TGF-β, BMP, FGF, IGF-1, and PTHrP are indeed significant, they are widely discussed in the literature, and their mechanisms overlap to some extent. Consequently, we have chosen to address them more concisely in the text. Finally, in response to the reviewers' comments regarding the direct relevance of biomaterials used for in vitro differentiation of mesenchymal stem cells (MSCs) or maintaining chondrocyte phenotype to osteoarthritis treatment in Section 4.3, we have further supplemented the discussion using 3D bioprinting, a cutting-edge technology in tissue engineering, as an example. The added content is highlighted in blue font in the "3D Bioprinting for Cartilage Injury Repair" section.
- The authors are strongly advised to conduct a thorough review of the manuscript to address grammatical and formatting issues.
A: We appreciate the reviewers' suggestions. We have conducted a thorough review of the manuscript and used ChatGPT to address the grammar and formatting issues, ensuring that the manuscript is more standardized and accurate in terms of language and formatting.
Round 2
Reviewer 2 Report
Comments and Suggestions for Authors
Although the author has made revisions to the manuscript, the changes do not adequately address the reviewers' concerns. The added content in the manuscript lacks a strong connection to the main subject of the article and fails to significantly improve its overall quality. While the paper discusses the relationship between chondrocyte hypertrophy and arthritis, it does not clearly explain why studying chondrocyte hypertrophy is of critical importance for arthritis treatment. Additionally, the content introduced in the latter part of the manuscript, though related to arthritis treatment, has limited relevance to chondrocyte hypertrophy. Furthermore, the author has not thoroughly reviewed the draft, as evident by errors such as the duplication of references 53 and 54, which remain uncorrected. Therefore, the reviewer does not recommend the manuscript for publication in this journal.
Author Response
Reviewer #2
Although the author has made revisions to the manuscript, the changes do not adequately address the reviewers' concerns. The added content in the manuscript lacks a strong connection to the main subject of the article and fails to significantly improve its overall quality. While the paper discusses the relationship between chondrocyte hypertrophy and arthritis, it does not clearly explain why studying chondrocyte hypertrophy is of critical importance for arthritis treatment. Additionally, the content introduced in the latter part of the manuscript, though related to arthritis treatment, has limited relevance to chondrocyte hypertrophy. Furthermore, the author has not thoroughly reviewed the draft, as evident by errors such as the duplication of references 53 and 54, which remain uncorrected. Therefore, the reviewer does not recommend the manuscript for publication in this journal.
- The added content in the manuscript lacks a strong connection to the main subject of the article and fails to significantly improve its overall quality. While the paper discusses the relationship between chondrocyte hypertrophy and arthritis, it does not clearly explain why studying chondrocyte hypertrophy is of critical importance for arthritis treatment.
A: What specifically do the additional contents refer to? We have addressed each of the reviewers' comments in detail. For example, the reviewer mentioned that the manuscript failed to clearly explain why studying chondrocyte hypertrophy is critically important for osteoarthritis (OA) treatment. In response, we referenced numerous review articles, such as Cells. 2022, 11, 4034; Bioengineering (Basel). 2023, 10, 741; Pharmaceutics, 2021, 13, 1139; Curr Rheumatol Rep. 2019, 21, 38; Tissue Eng Part A, 25, 1369; Med Sci (Paris). 2018, 34, 1092; Oncotarget. 2017, 8, 91316; J Cell Physiol. 2018, 233, 1940; Int J Mol Sci. 2015, 16, 19225; Genes Dis. 2015, 2, 76, which demonstrate the close association between chondrocyte hypertrophy and osteoarthritis.
As chondrocyte hypertrophy progresses, various biological markers associated with OA are expressed. For example, MMP-13 promotes the degradation of cartilage matrix, compromising cartilage integrity; ALP, type I collagen, and angiogenic factors contribute to and exacerbate pathological bone formation. The increased presence of these biological markers is a direct driver of OA progression. As the reviewer noted, while the precise pathogenesis of OA is not yet fully understood, type X collagen can serve as a therapeutic target. If chondrocyte hypertrophy can be effectively controlled, the expression levels of MMP-13, ALP, type I collagen, and angiogenic factors would decrease significantly, thereby naturally mitigating OA progression.
This highlights the critical importance of controlling chondrocyte hypertrophy for OA treatment. The additional content mentioned above serves as a foundation to support the rationale for "targeting chondrocyte hypertrophy as a strategy for OA treatment," further emphasizing the therapeutic potential of addressing chondrocyte hypertrophy. As for the reviewer’s comment that the manuscript “did not significantly enhance the overall quality of the article,” we believe the opposite is true. These supplementary contents strengthen the scientific argument of the manuscript and further underscore the feasibility and importance of controlling chondrocyte hypertrophy as a therapeutic strategy for OA.
- The content introduced in the latter part of the manuscript, though related to arthritis treatment, has limited relevance to chondrocyte hypertrophy.
A: In the tissue engineering strategies for suppressing chondrocyte hypertrophy, we first explored how the three essential elements of tissue engineering—biomaterials, cells, and bioactive factors—can individually control the occurrence of chondrocyte hypertrophy. This approach stems from the fact that, in the practical application of tissue engineering, it is common to flexibly select a single factor, a combination of two factors, or all three factors together based on specific clinical needs and on-site conditions to address real-world problems such as arthritis.
As the reviewer pointed out, the lack of in vivo evidence in this section is primarily due to the limited number of relevant studies, which falls short of meeting the reviewer's expectations. However, the addition of this content was intended to address the suggestions made by another reviewer. Although the reviewer noted that "the added content, while related to arthritis treatment, has limited relevance to chondrocyte hypertrophy," this supplementary material, when combined with the earlier discussion of the three essential elements of bioengineering, can provide new insights for researchers in the field of osteoarthritis. Specifically, integrating these three essential elements with advanced tissue engineering technologies, such as in situ bioprinting, can yield tissue-engineered products that better meet clinical needs. This approach also holds promise for addressing the lack of literature highlighted by the reviewer by opening new directions and methods for research. Such integration not only enriches the depth of the study but also contributes to advancing osteoarthritis treatment technologies further. Finally, to respect the reviewer's opinion, we have incorporated the only in vivo study related to chondrocyte hypertrophy we could find into Section 4, “Tissue Engineering Methods for Inhibiting Chondrocyte Hypertrophy,” specifically under subsection 4.3, “Biomaterials.” This addition has been highlighted in green text for the reviewer’s reference.
- 3. The author has not thoroughly reviewed the draft, as evident by errors such as the duplication of references 53 and 54, which remain uncorrected. Therefore, the reviewer does not recommend the manuscript for publication in this journal.
A: We acknowledge that we made a significant mistake in organizing the literature. Due to the large volume of references, we utilized reference management software to minimize errors during the process. However, despite our efforts, a major oversight occurred during the final review. We have since corrected the issue. This experience has made us deeply aware of the shortcomings in our work, and we are determined to adopt a more diligent and meticulous approach in the future to prevent similar situations from happening again.